# The Suitability of Latex Particles to Evaluate Critical Process Parameters in Steric Exclusion Chromatography

**DOI:** 10.3390/membranes12050488

**Published:** 2022-04-30

**Authors:** Friederike Eilts, Marleen Steger, Keven Lothert, Michael W. Wolff

**Affiliations:** 1Institute of Bioprocess Engineering and Pharmaceutical Technology, University of Applied Sciences Mittelhessen (THM), Wiesenstr. 14, 35390 Giessen, Germany; friederike.eilts@lse.thm.de (F.E.); anna.marleen.steger@lse.thm.de (M.S.); keven.lothert@lse.thm.de (K.L.); 2Fraunhofer Institute for Molecular Biology and Applied Ecology (IME), Ohlebergsweg 12, 35392 Giessen, Germany

**Keywords:** aggregation, crowding out, downstream processing, pH, polyethylene glycol, SXC, precipitation

## Abstract

The steric exclusion chromatography (SXC) is a rather new method for the purification of large biomolecules and biological nanoparticles based on the principles of precipitation. The mutual steric exclusion of a nonionic organic polymer, i.e., polyethylene glycol (PEG), induces target precipitation and leads to their retention on the chromatographic stationary phase. In this work, we investigated the application of latex particles in the SXC by altering the particle’s surface charge as well as the PEG concentration and correlated both with their aggregation behavior. The parameters of interest were offline precipitation kinetics, the product recovery and yield, and the chromatographic column blockage. Sulfated and hydroxylated polystyrene particles were first characterized concerning their aggregation behavior and charge in the presence of PEG and different pH conditions. Subsequently, the SXC performance was evaluated based on the preliminary tests. The studies showed (1) that the SXC process with latex particles was limited by aggregation and pore blockage, while (2) not the aggregate size itself, but rather the aggregation kinetics dominated the recoveries, and (3) functionalized polystyrene particles were only suitable to a limited extent to represent biological nanoparticles of comparable size and charge.

## 1. Introduction

The steric exclusion chromatography (SXC), as an upcoming purification technique for biological macromolecules and viruses, was first introduced by the work of Lee et al. [1] in 2012. The authors described the method to be based on the molecule capture at a hydrophilic surface without direct chemical interactions, but induced by mutual steric exclusion. The working principle of the SXC is closely related to precipitation, and extensively described elsewhere [1,2]. Here, only a summary of the core concepts relevant for this work is presented, and their impact on the SXC operation. This is visualized schematically in Figure 1. First, the SXC is often operated with polyethylene glycol (PEG), which serves as the driving force for strong mutual steric exclusion of chemically non-reactive solutes from one another [3,4]. This process leads to a deficient zone around given solute molecules, e.g., virus particles. Here, the concentration of the crowding solutes, in this case PEG, is lower than in the bulk solution [5,6]. The deficiency is caused by two phenomena [2]: On the one hand, the deficient zone is incompletely filled by the PEG spheres due to the fact that two solid objects cannot exist in the same place at the same time. On the other hand, the water molecules, which are smaller in size than the PEG, occupy the free space by preferential inclusion, consequently, leading to a lower PEG concentration in the PEG-deficient zone. The presence of the PEG-deficient (around virus particles and the stationary chromatographic phase) and PEG-concentrated (bulk solution) zones causes an unfavorable increase in free energy (Figure 1A, Load). Hence, the system strives for a thermodynamic stability, realized by depletion effects [7], which are defined by the solubility of the solutes [8]. This implies that solvent (water molecules) from the PEG-deficient zones is released by the association of the virus particles and their accretion at a surface, e.g., the stationary phase. Hence, the target viruses arrange themselves on the chromatographic material, while contaminants may be washed out (Figure 1A, Wash). To reverse the binding, the PEG content is reduced, or omitted, and the virus particles are eluted in a concentrated, purified state (Figure 1A, Elute).

The selective interaction of virus particles depends on several factors related to their characteristics and the solution composition. The two most important ones for this work are (1) the size and concentration of the solutes, which determine the extent of the mutual steric exclusion. Thus, the smaller and the lower concentrated the virus particles are, the higher concentrated and the higher the molecular weight of the PEG must be, and vice versa [4,5,9]. (2) Accordingly, the chemical surface characteristics of the virus particles determine repulsive and attractive forces and can therefore prevent accretion [6,10,11]. This effect may be augmented by changes in pH, temperature, and salt composition, all having a direct impact on the electrostatic behavior and solubility [1,2,12].

For the operation of the SXC, considerations of practical relevance include the use of a liquid chromatographic (LC) system. As explained, the SXC is based on precipitation of the targets with retention at the stationary phase. Thus, the process is also limited by the target association itself, which can cause pore blockage of the adsorbent, if either the capacity of the column is reached, or bigger precipitates are formed throughout loading. Therefore, pressure limits are of utmost importance for the stationary phase as well as for the LC system. In the past, convective media, such as monoliths [1,2,13,14] and membranes [15,16,17,18,19], were applied to counteract diffusion limitations and clogging caused by the virus particles’ large size. As a secondary effect, the high viscosity of the PEG solutions was easier to handle with these stationary phases. Last, in-line mixing, representing the blending in the mixer and tubes of the LC immediately before loading onto the stationary phase, was proposed to reduce a possible pressure increase [1,18], but several authors worked on a laboratory scale with off-line mixing without describing such limitations [10,15,16,17,18].

As concluded from these concepts, the role of charge-induced and pre-column aggregation has not yet been discussed. We aimed to tackle this question with an artificial particle system. Certainly, the latter remains to be validated with corresponding biological nanoparticles, e.g., viruses; nevertheless, the advantages concerning the continuity of the system prevailed. More precisely, viruses are heterogeneous particles with batch-to-batch variances. Additionally, virus inactivation may impede quantification. Therefore, in this work, polystyrene calibration particles (CP) were selected, as they are stable in harsh environmental conditions, i.e., degradation was unexpected, and recovery losses may be attributed to retention on the stationary phase. Furthermore, CP of varying sizes are easily available with different functionalization and have previously been used for chromatographic modeling [20,21]. This article reports first results from the experimental evaluation of the suitability of latex particles to identify critical process parameters of SXC applications, and the role of target aggregation in the process.

## 2. Materials and Methods

All chemicals were purchased from Carl Roth (Karlsruhe, Germany) if not indicated otherwise.

### 2.1. Polystyrene Calibration Particles Characterization

CP particles with 190 nm mean diameter, were purchased with two types of surface functionalization, either sulfate-or hydroxyl-groups, CPS and CPH, respectively (Polybead, Polysciences Europe, Hirschberg a. d. Bergstrasse, Germany). Size distribution and charge measurements were performed with a Zetasizer Nano ZS90 and the corresponding Zetasizer software (version 7.13) (both Malvern Panalytical, Malvern, UK) according to Lothert et al. [16] with the deviation of the pre-set “polystyrene latex”. CP with a final concentration of 10^10^ particles mL^−1^ were prepared in 0.1 M citrate phosphate buffer (CPB), the conductivity (15 mS cm^−1^) adjusted with NaCl. Depending on the experiment, the CPB was of varying pH (3.0–7.4), PEG_8000_ concentration (0–10% (*w*/*v*)), 8000 Da molecular weight, and polysorbate 20 (Tween 20) addition (0% or 0.03%). Every sample preparation was carried out directly before analysis, including 2 min ultrasonication and 30 s mixing by vortex. Kinetics were recorded automatically with a measurement every 5 min.

### 2.2. Steric Exclusion Chromatography

All SXC experiments were conducted on an Äkta Pure 25 (Cytiva, Marlborough, MA, USA) with online UV (260 nm), pre-column pressure, and dynamic light scattering (DLS) (Nano DLS Particle Size Analyzer, Brookhaven Instruments, Holtsville, NY, USA) monitoring, controlled via the Unicorn 7.1 software (Cytiva, Marlborough, MA, USA). The execution of the SXC runs was adapted from Lothert et al. [17] with minor adjustments. In short, the single-use adsorbent was a stack of ten layers of regenerated cellulose membranes (Whatman, Maidstone, UK), 13 mm in diameter and 1 µm nominal pore size, inserted into a stainless-steel filter holder (Pall Life Sciences, Port Washington, NY, USA). Before assembly, the membranes were left for swelling in PBS overnight. The volume of the stationary phase was 0.1 mL. However, the effective column volume was 1.2 mL, including connectors and fittings. The prepared membranes were equilibrated with 10 mL CPB of the desired pH and PEG_8000_ concentration. For the batch loading (10 mL), the CP were diluted in the respective equilibration buffer aiming for a concentration of 10^10^ particles mL^−1^, filled into a 10 mL superloop, and loaded onto the column. Afterwards, no washing step was performed as no contaminants were present, but the CP were eluted with 20 mL CPB, always at pH 7.4, without PEG_8000_, but supplemented with 0.03% polysorbate 20. The flow rate for all steps was set to 0.5 mL min^−1^. The quantification of the CP was performed indirectly via measurement of the optical density at 260 nm. The relative recoveries were determined based on the concentration of the initial load sample. All experiments were carried out in triplicates.

## 3. Results

### 3.1. Polystyrene Particle Characterization

For comparative reasons, the CP were chosen to be spherical and of an average size (190 nm) of viruses, which were successfully purified by the SXC in previous studies [15,17,19]. In order to characterize the CP aggregation behavior, the pH-and PEG_8000_-dependent size and charge distributions were assessed. Their concentration was chosen according to previous size and charge measurements, indicating a concentration-independent plateau for 10^9^–10^11^ particles mL^−1^ (data not shown). Firstly, CPH and CPS samples were titrated to pH 3.0–7.4 without the addition of PEG_8000_ (Figure 2A,B). Concerning the pH-dependent size (Figure 2A), CPS revealed constant values of 231 ± 3 nm, also in the presence of polysorbate 20 (0.03%). Likewise, the obtained size of CPH behaved independently of the pH. However, the mean value of 721 ± 105 nm without polysorbate 20 addition, and 300 nm in its presence (0.03%), was larger than the diameter stated by the manufacturer in PBS at pH 7.4 (190 nm) as well as the diameter of the CPS. It should be noted that the diameters for CPS and CPH in a second phosphate buffer, PBS, with and without polysorbate 20, was around 200 nm (data not shown). Next, the CP charge varied in a pH-dependent manner (Figure 2B). The zeta potential of CPS and CPH similarly increased with reduced pH values, but CPH revealed higher zeta potentials. CPS samples exhibited the lowest charge (−34 mV) at pH 7.4, and the highest (−13 mV) at pH 3.0. For CPH, −19 mV was measured at pH 7.4, and −6 mV at pH 3.0. Additionally, only negative charges were obtained for the tested pH range. Hence, no isoelectric point (pI) could be determined. Last, polysorbate 20 increased the zeta potential at pH 3.0 and pH 7.4 to −3–−7 mV, respectively.

Secondly, aggregation kinetics were performed to clarify possible interactions of the two parameters, pH and PEG_8000_ concentration. Four selected pH/PEG combinations were chosen, to match conditions applied for similar-sized viruses [15,17], and to represent extremes for more pronounced results. The obtained kinetics showed positive linear trends with mostly steeper slopes (*a*) for bigger initial precipitate sizes (Figure 2E,F). The only exceptions were CPS with the conditions 8% PEG_8000_ at pH 7.4, and CPH with 2% PEG_8000_ at pH 7.4. The R^2^ for the fits were above 0.91 for all data sets, apart from the recording of CPH with 8% PEG_8000_, pH 3.0 (0.53), which also had a reduced a-value for the slope. In comparison, all tested conditions indicated bigger sizes for CPH than for CPS at *t* = 0 min as well as after 2 h of incubation time, except for pH 3.0 with 2% PEG_8000_. The latter showed similar end values of roughly 2000 nm. The biggest sizes were reached at pH 3.0 (8% PEG_8000_), 3000 nm (CPS) and 6000 nm (CPH). In detail, the aggregation kinetics of CPH were controlled by the PEG_8000_ concentration. At 2%, similar size distributions were measured for both pH values, but the slope was steeper for pH 3.0 (*a* = 0.918) as compared to pH 7.4 (*a* = 0.519) (Figure 2E). An increase to 8% PEG_8000_ led to significantly elevated mean sizes of 4000 nm and 6000 nm after 2 h for pH 7.4 and pH 3.0, respectively, but with comparable slopes of the fits of 1.104 and 0.924 (Figure 2C,E). Next, the size measurements of CPS revealed a pH-dominant distribution (Figure 2D,F). After the full incubation time, 1000 nm were measured for pH 7.4 with 2% PEG_8000_, followed by similar trends for pH 7.4 combined with 8% and pH 3.0 with 2% PEG_8000_ (1500 nm). Concerning the latter two combinations, the slope was flatter for pH 7.4 (*a* = 0.438) compared to pH 3.0 (*a* = 0.740).

### 3.2. Steric Exclusion Chromatography with Polystyrene Particles

To evaluate the role of CP aggregation for the SXC, the parameters were chosen to correspond to the particle aggregation kinetics (Figure 2C–F). First focusing on the CPH, processing by SXC revealed 15% or less recovery of particles in the two fractions, flow-through and elution combined, for all combinations tested (Figure 3A). More specifically, only for the loading conditions pH 7.4 with 2% PEG_8000_, CPH were recovered in the elution fraction with approximately 8% of the total load. Concerning the flow-through, both applications of pH 7.4 resulted in the detection of 7%, and the pH 3.0 runs in 2% of the initial particle load. These observations were supported by the chromatograms (Figure 4A–D). No changes in the UV signal were recorded over the full process length, except for pH 7.4 with 2% PEG_8000_ (400 mAU in the elution peak) (Figure 4A).

A similar behavior was obtained for the CPS applications at pH 3.0, where the lower PEG_8000_ concentration (2%) led to a small CPS yield in the elution fraction (11%), while 8% PEG_8000_ inhibited the particle passage in flow-through and elution. The application of pH 7.4 revealed approximately 90% and 60% of total recovery for 2% and 8% PEG_8000_, respectively. Here, the lower PEG_8000_ concentration of 2% resulted in a yield of 21% CPS in the elution fraction, and the higher concentration showed 8% yield (Figure 3B). Again, the results were supported by the UV signals (Figure 4E–H), indicating elution peaks, 100 (G), 700 (F), and 800 mAU (E) corresponding to increasing yields and recoveries. Additionally, the presence of CPS in the flow-through fractions (51% and 71%) at pH 7.4 were suggested by break-through UV signals of 150–200 mAU (Figure 4E,F).

Lastly, the pre-column pressure was monitored online to examine putative particle adhesion to the chromatographic membranes and pore blockage of the same. The pressure was increased for all runs throughout the loading process (Figure 4). Here, the initial pressure and the pressure increase were higher for elevated PEG_8000_ concentrations as well as for reduced pH values. Furthermore, CPH runs revealed higher final pressures than CPS ones.

## 4. Discussion

This study aimed to investigate whether functionalized CP, CPS and CPH, are a potential option to analyze critical process parameters of the SXC. Additionally, correlations of their pH- and PEG-dependent precipitation behavior and the SXC yields and recoveries were intended to elucidate pore blockage events and retention patterns. Therefore, the particles were characterized, and the precipitation kinetics recorded. The applied parameters were used in the SXC process afterwards.

### 4.1. Particle Characterization

Firstly, the pH-dependent size measurements showed constant values for CPS (230 nm) and CPH (720 nm) in CPB. The addition of polysorbate 20 revealed mean values of 300 nm for CPH. Presumably, the addition of polysorbate 20 helped to reduce spontaneous aggregation of CPH in the presence of CPB. Interestingly, the values in PBS for CPS were around 200 nm, comparable to the manufacturer specifications (190 nm) (Figure 2A). All measurements were done using a dynamic light scattering system. Thus, we attributed the deviations, to differences in the hydration shell in the presence of CPB and polysorbate 20, which influenced the Brownian motion, i.e., the recorded parameter in DLS measurements. Kinetic studies with different buffers and 0% PEG could clarify this observation. However, we decided to use CPB as buffering substance nevertheless, although PBS seemed to increase particle stability on solution. The main reason for this decision was that PBS has no buffering capacity over the full range of the conducted experiments, whereas CPB covers pH 2.5–7.5. Secondly, no pI was observed for CPH or CPS (Figure 2A,B) in the pH range 3.0–7.4. According to the functional groups, the pI was expected at pH 1–2 for particles functionalized with sulfate groups [22] and pH 6–7 for hydroxy groups [23]. However, here, a decreasing pH, accompanied by a zeta potential increase, showed a comparable function for both, CPH and CPS. Accordingly, the exhibited charge could be attributed to the core latex material, which contained sulfate groups from manufacturing (communication with Polysciences Europe, Hirschberg a. d. Bergstrasse, Germany). Furthermore, the association of anions to the spheres themselves should contribute to the observed negative zeta potential values in the range down to pH 3. Anions have been reported to adsorb to hydrophobic surfaces and to deprotonate counterions, like carboxyl groups [24]. Thus, measurements of the zeta potential might not reflect the neutral state of a surface that is in interaction with an anion-containing solution. Nevertheless, the additional sulfate functionalization of the CPS led to an increased electrostatic repulsion compared to the CPH, indicated by the lower sizes and zeta potential values. The lower zeta potential of CPS than CPH in the presence of 0.03% polysorbate 20 supports this assumption. The exhibited surface charge of CPS was putatively less effectively shielded than that of CPH. We also assume that these differences in the electrostatic repulsion were the main influencing factor in the aggregation kinetics (Figure 2C–F). While the aggregation of CPH was enhanced by the addition of PEG_8000_, the CPS samples indicated the need for a reduction of the repulsion (by lowering the pH) to enable aggregation. This could be further investigated by the addition of high concentrated salts, which largely eliminates electrostatic interactions. Additionally, the use of different-sized CP and PEG could help to understand the role of their solubility in the system.

Last, the low value of the R^2^ for the fit of CPH with 8% PEG_8000_ at pH 3.0 (0.53) was presumably caused by sedimenting aggregates, due to the lack of agitation. This phenomenon was observed for the latex particles as well as different viruses, but not quantitatively recorded. Additionally, we assume that the non-spherical form of the aggregates caused deviating results, depending on the measured orientation, and this was more pronounced for the bigger sizes. The application of DLS for the recording of aggregation kinetics has its limitations, especially in a static particulate system, where sedimentation can alter the results. However, we expected to derive indications concerning the influence of pH and PEG_8000_ concentration from the kinetics, rather than the knowledge of the finite size. This point should be especially stressed as the SXC operation introduces shear forces, which might break-up aggregates [25] and cannot be reflected by the kinetics presented in this work.

### 4.2. Steric Exclusion Chromatography

Next, the four different conditions of PEG/pH-combinations, which were used for the aggregation kinetics, were now applied in the SXC. Both particle types revealed very low yields in the elution fractions of ≤21% for CPS and ≤7% for CPH, either due to low total recoveries, or due to a breakthrough during loading (Figure 3). None of the tested conditions were appropriate to cause particle retention throughout loading and allow for a full elution at the same time. These results were contradictory to previous applications of the SXC with 8% PEG_8000_ (pH 7.4) for a similar-sized virus, the Orf virus, which revealed >90% yield [17]. A similar zeta potential of 15 mV with the same PEG_8000_ concentration showed virtually no recovery of the CP. Thus, we assumed the recovery losses were caused by dense particle aggregates, which remained on the stationary phase and induced filtration effects. This aspect will be discussed later in more detail. Additionally, an incubation of the loaded column, with salt- or surfactant-containing buffer throughout the elution [15], could improve the elution conditions and would help to understand the incomplete mass balances. Another explanation for the low recoveries could be (un-)specific binding to the membranes. However, we excluded this hypothesis at least for the CPS at neutral process conditions, as substantial amounts of the beads were obtained in the flow-through fractions.

The occurrence of CPS in the flow-through fraction was diminished by increased PEG_8000_ concentrations and reduced pH values. For these conditions, the slope of the pressure increase was considerably steeper than reported for biological nanoparticles using the SXC [1,10,16,17]. We propose two reasons for this behavior: (1) the approximated two-dimensional surface charge distribution of the hard latex spheres is less complex than the three-dimensional characteristics of so-called soft particles, e.g., viruses [26,27,28,29,30], which additionally have an amphoteric character. Furthermore, the distribution of the functional groups is assumed to be more homogenous on chemically defined hard particles. These characteristics cause more compact aggregates with reduced permeability [11]. Further investigations might focus on the addition of salts to investigate the reduction of electrostatic attraction [1], and the impact of the functionalization on the precipitation. (2) The high concentration and purity of the latex suspensions led to increased crowding-out effects as no other solutes interfered with the osmotic force induced by the PEG_8000_ [1]. Lowering the particle concentration (Figure 1B) was no option, as unstable size and charge measurements were obtained in the pre-experiments. Therefore, reduced PEG concentrations might improve a comparison of the latex particle system with the biological counterparts. We would further like to point out that latex particles are presumably more hydrophobic [31] than viruses [32]. This characteristic should lead to reduced SXC efficiencies. On the one hand, the binding to the hydrophilic cellulose membranes is expected to be reduced, and on the other hand, PEG is less effectively excluded from hydrophobic surfaces, having a slightly hydrophobic character itself [1].

It should be mentioned that, although the exact values, i.e., the charge of similar-sized particles, may not be appropriate to compare the SXC applications, general core concepts could be extracted, e.g., the pre-column pressure behavior throughout loading. The initial pressure was dominated by the PEG_8000_ concentration, drastically increasing for loading conditions with reduced pH values. As mentioned, we attributed this to the formation of big aggregates causing pore blockage. As a secondary effect, increased PEG_8000_ concentrations cause higher viscosities, reinforcing the behavior. For the purification of a hepatitis C virus by SXC, a similar trend was reported [10]. Here, the approximation of the pH to the pI caused increased yields and higher pre-column pressures throughout loading, i.e., the reduced repulsion caused stronger accretion. However, the limitations of this concept, and the impact of the aggregate size and aggregation kinetics on the SXC, have not yet been evaluated. Presumably, the identification of an operation window allows for a stable SXC operation without pore blockage events. Thus, the optimization of the SXC operation itself is more complex than traditional PEG precipitation, as concerns of elevated precipitation are not existent for the latter. Thus, optima for a sufficient target association without instant pore blockage should be investigated.

For the time-interval of loading (20 min), only the aggregate sizes equaling the pore diameter of the stationary phase (1 µm) (Figure 2E,F) revealed recoveries in the elution fraction (Figure 3). Presumably, stationary phases with bigger pore sizes might help to overcome the observed limitations. Future experiments with different pore structures will give valuable insights into the correlation between target precipitation, target retention, and pore blockage. However, it should be noted again, that the aggregation kinetics in this study were of a static character. Thus, they were not representative for the true precipitate size that was loaded onto the stationary phase, but rather an indication of the aggregation behavior. For predicting the particle retention, these statically measured aggregate sizes did not correlate with the recovery in the flow-through fraction. This was especially visible for the CPS (Figure 2F and Figure 4B). In more detail, the aggregate size and therefore the retention, due to filtration effects, can explain the higher total recoveries of CPS for 2% PEG_8000_ compared to 8% at pH 7.4, but not the zero breakthrough for 2% PEG_8000_ at pH 3.0. In this case, the steeper slope of the aggregation kinetics of the latter might explain the increased retention. The steepness represents the likelihood of bead association and retention. For all observed cases, the particle breakthrough was reduced with an increasing slope of the kinetics. However, the effects were not transferable between the two different latex particle types by sheer number.

We further concluded that the leveling pre-column pressure, following particle breakthrough of the CPS during loading at pH 7.4 (Figure 4E,F), was caused by repulsion, impeding a stable adherence to layers of particles inside the chromatographic membrane. More precisely, the CPS accreted to the hydrophilic stationary phase throughout loading due to crowding-out effects. At a certain point, the membrane pores were covered with CPS, which exhibited a high charge repulsion at this pH (−34 mV). This prevented further layering. A similar effect was observed for the phage M13K07 [1], a linear, uniformly negative-charged nanoparticle. Thus, the charge distribution on the latex particle surface could be the cause of this behavior. As already suggested, the addition of high salt concentrations would help to differentiate between the effects of the electrostatic interaction and the solubility of the beads throughout this process.

In summary, next to the already stated open questions, a study of PEG concentrations and pH values between the tested ranges would result in a further understanding of the retention mechanisms throughout the SXC process with CP. Nevertheless, we propose a general suitability of CPS for SXC description in the context of the tested conditions, as both extrema of full retention and complete breakthrough were observed for the CPS. Concerning an imitation of virus processing by SXC, we assume that further experiments with protein-functionalized CP in a wide pH and salt concentration range would be required. Such a study could be augmented by the inclusion of varying column-loading concepts. In our study, bulk loading was performed, but continuous in-line mixing could reduce variations of the holding time in the superloop, and improve column characterization.

## 5. Conclusions

In the present study, the suitability of polystyrene particles, CPH and CPS, to determine critical process parameters for the SXC process, was investigated. We assume that the recoveries and yields were controlled by the attractive and repulsive forces of the particles, determining their aggregation kinetics, i.e., size distribution and slope. However, the role of solubility, in contrast to electrostatic interaction, was not investigated. More precisely, the aggregate size served as an indicator for the total recovery in most cases, which was reduced with increasing size. Additionally, for similar sizes, an increase of the kinetics’ slopes also indicated reduced total recoveries. Especially for CPS, the slope as well as the mean aggregate size predicted relative total recoveries and yields. Overall, the highest yields were achieved for the lowest slope values for both particle types. This observation served as a general concept, but the exact numbers were not transferable between the CP. In conclusion, the determination of aggregation kinetics were helpful for understanding losses throughout the SXC process, describing relative yields and recoveries. We propose this first description of the SXC by the target aggregation behavior as helpful for future investigations of the fundamentals of this method.

## Figures and Tables

**Figure 1 membranes-12-00488-f001:**
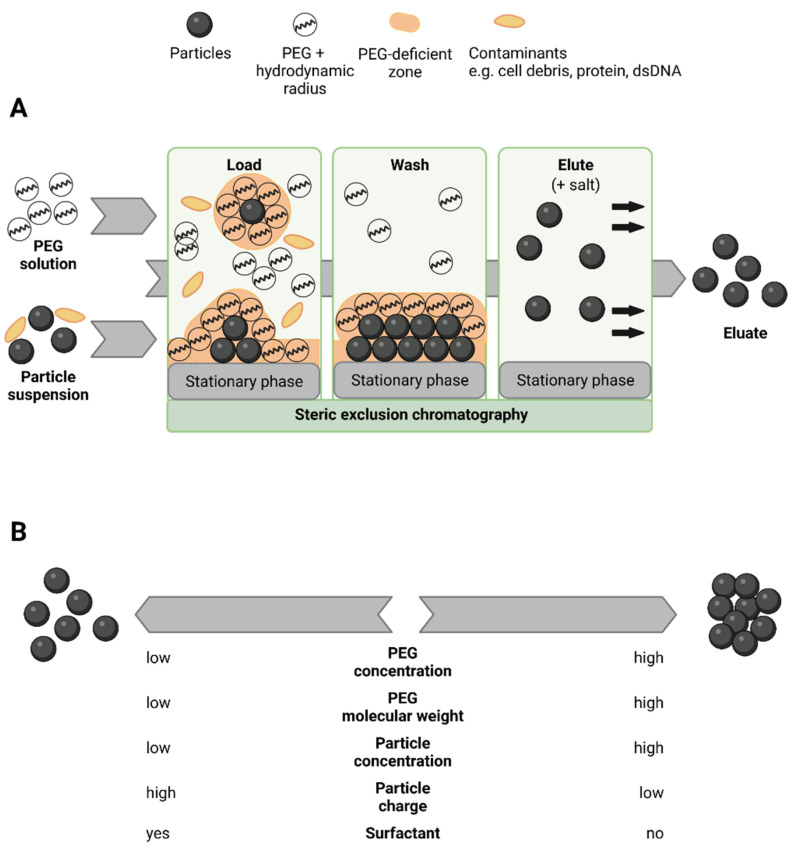
Theory of working principle and influencing factors of the steric exclusion chromatography (SXC). The SXC is a chromatographic method based on the core-concepts of polyethylene glycol (PEG) precipitation: (**A**) represents the steps of the method’s application as a chromatographic purification technique. (**B**) depicts a selection of parameters influencing the target aggregation and accretion behavior, which were found to be critical for the SXC. Created with bioRender.com.

**Figure 2 membranes-12-00488-f002:**
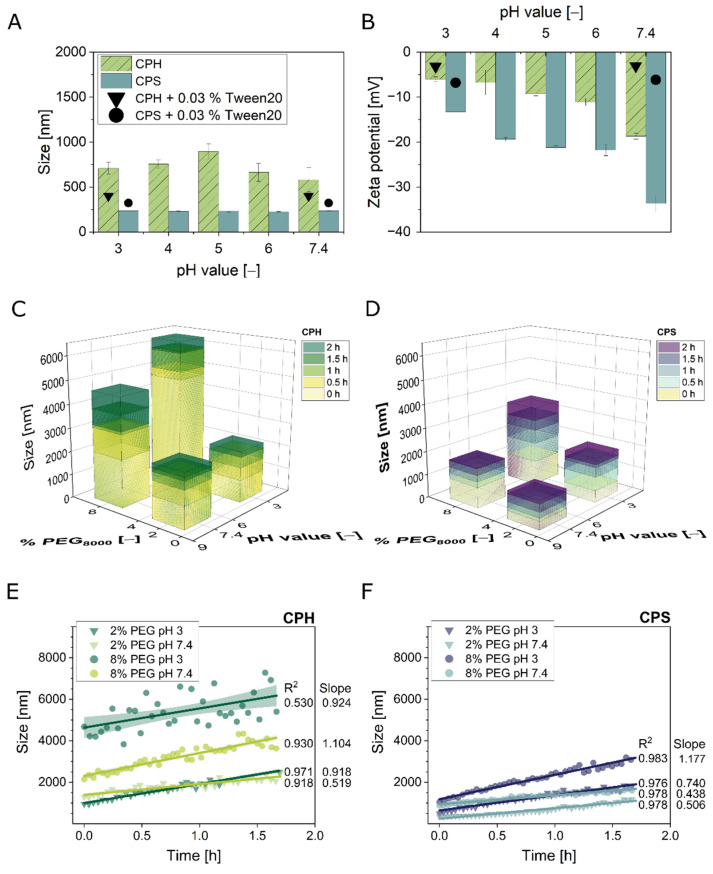
Polystyrene particle characterization. The functionalized polystyrene particles, sulfated (CPS) and hydroxylated (CPH), were characterized prior to the chromatographic experiments. (**A**,**B**) show the size distribution and electrophoretic mobility, expressed as zeta potential, of CPH (green, striped) and CPS (blue) at varying pH (and 0% PEG_8000_). Additionally, the particles were supplemented with 0.03% polysorbate 20 (Tween 20) (black triangle and circle, respectively). Visualized are the measurements directly after the sample preparation at *t* = 0 min. The data shows the mean of *n* = 3 and standard deviations as error bars. (**C**–**F**) present the size distribution kinetics of the pH- and PEG_8000_-dependent particle aggregation behavior analyzed over a period of 2 h. The data shows the means of *n* = 2. (**E**,**F**) depict linear fits for the presented kinetics, including all generated data points, with the respective *R*^2^ and slopes. The lines represent the fits, while the hatched areas indicate the 95% confidence intervals. Data in (**E**,**F**) partly overlaps with the data shown in (**C**,**D**), but is included to facilitate the overview.

**Figure 3 membranes-12-00488-f003:**
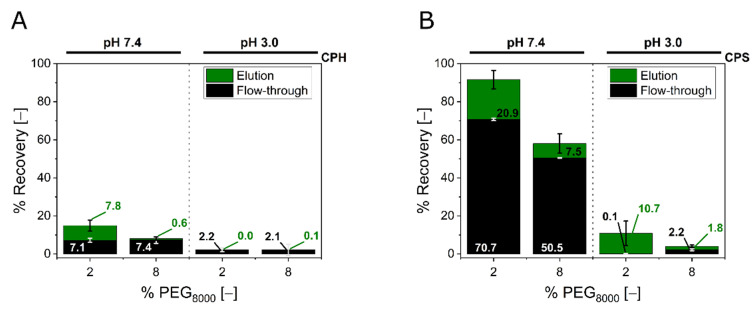
Recoveries of steric exclusion chromatography (SXC) runs. (**A**) represents the particle recoveries of SXC runs with the hydroxylated particles (CPH), chromatograms visualized in Figure 4A–D, while (**B**) accounts for sulfated beads (CPS) from Figure 4E–H. The recoveries of the different fractions were normalized to the respective initial load sample, accounting for volumetric ratios. Hence, the cumulative load is 100%, while elution (black), flow-through (green), and losses (not depicted) sum-up to this number. The CP concentration was indirectly determined by optical density at 260 nm. The bars represent means of triplicate runs and their respective standard deviations.

**Figure 4 membranes-12-00488-f004:**
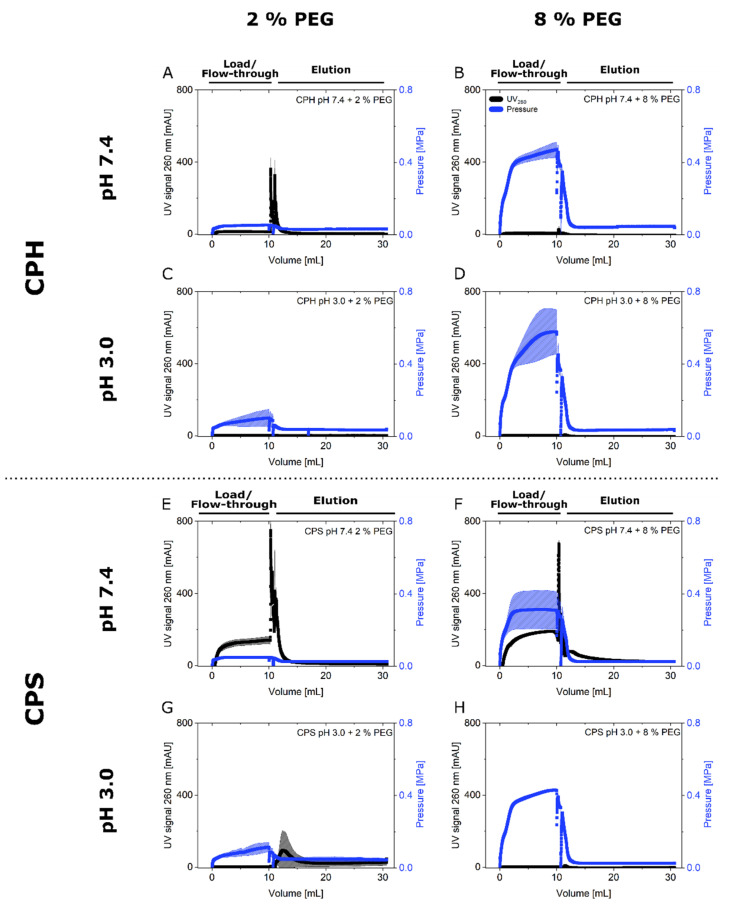
Chromatograms of steric exclusion chromatography (SXC) application. Hydroxylated (CPH) and sulfated (CPS) polystyrene particles were processed via SXC. For each experiment, 10 mL of the pre-mixed suspension were loaded onto the column. Elution (20 mL) was performed at neutral pH 7.4 (0% PEG_8000_ and 0.03% polysorbate 20). The UV signal at 260 nm (black) and the pre-column pressure (blue) were recorded online. The solid lines represent the means of t *n* = 3 runs, whereas the hatched areas indicate the standard deviations. The load of the experiments with CPH (**A**–**D**) was modified with 2% (**A**,**C**) or 8% (**B**,**D**) PEG_8000_. Additionally, the pH was varied between 7.4 (**A**,**B**) and 3.0 (**C**,**D**). Like the CPH experiments, CPS (**E**–**H**) loads contained 2% (**E**,**G**) or 8% (**F**,**H**) PEG_8000_ and the pH was varied between 7.4 (**E**,**F**) and 3.0 (**G**,**H**).

## Data Availability

All relevant data are within the paper.

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
