# Peer review of "The Suitability of Latex Particles to Evaluate Critical Process Parameters in Steric Exclusion Chromatography"

_membranes, 2022, doi:10.3390/membranes12050488_

Round 1

Reviewer 1 Report

This paper was a pleasure to read with a good story presented in a clear manner. The authors talking about development of surrogate latex particles to be used as a surrogate for a new method of purification known as Steric Exclusion Chromatography. This method isn't very common in the biopharmaceutical industry, but could be relevant for process development with emerging modalities. I believe this paper is acceptable for publication provided the following comments are addressed:

  1. Pre-column pressures (L222) - regardless of the particle, I would not read too much into this because pre-C pressures depend on a lot of things. If this is indeed important from a scale-up perspective, then I would discuss the implication briefly.
  2. L264 to end of the paragraph - seems like somewhat poor R2 for CPH with 8% PEG at pH 3 needs to be supported with some evidence (prior work) if possible, since it seems mostly speculative - which is reasonable, but without any evidence directly, it is hard to see.

Reviewer 2 Report

This paper examined the use of latex particles with different surface functionalization as a model system to study the steric exclusion chromatography of viruses and other large particle therapeutics. Overall, the paper is well-written – I had just a few minor corrections regarding word choice.  The paper would have been much stronger if the authors could have included side-by-side comparison of the behavior of the latex particles with that of a virus, although I do appreciate the challenge in performing the virus experiments.  The results do provide some useful insights into the effects of various process parameters on the performance of steric exclusion chromatography with particles that are around 200 nm in size.  I think the paper could be published if the authors are able to effectively address the following concerns:

  1. Why did the authors use a membrane with 1 µm pore size when the measured size of the latex particles was as large as 6 µm under some conditions? Even the initial size of the CPH after mixing with PEG was close to 5 µm in the 8% PEG solution at pH 3.  It is not at all surprising that these large particles blocked the pores of the membrane, leading to a significant increase in pressure and low particle recovery.  Were any attempts made to use membranes with larger pore size?

  1. The data in Figure 2 show that the addition of Tween 20 reduces the size of the CPH particles at both pH 3 and 7 but increases the size of the CPS particles. What is the origin of this difference, particularly given that the addition of Tween seems to have a similar effect on the zeta potential for both particles?

  1. The authors hypothesized that the greater pressure increase for the latex particles compared to viruses was due to either the less complex charge distribution or to the high concentration and purity of the latex particles. Although both of these are possible, I would also argue that the surface of the latex particle is considerably more hydrophobic than the surface of a virus. These differences in hydrophobicity could easily lead to greater aggregation / precipitation and stronger binding to the membrane. This should be discussed in the paper.

  1. The authors commented that “the diameters for CPS and CPH in a second phosphate buffer, PBS, with and without polysorbate 20, was around 200 nm (data not shown).” Why weren’t any experiments with PEG performed in the PBS given that the CP seem to be much more stable in that buffer (at least based on the size measurement and its agreement with the expected size of these latex particles)?

  1. The authors attributed the negative charge on the CPH particles to sulfate groups in the core latex. Although this may well have contributed to the charge, almost all particles preferentially “bind” negative ions from solution due to the different hydration of the anions and cations in water. This “anion association” gives surfaces a negative charge (as determined by zeta potential measurements), even for surfaces that appear electrically neutral. I would encourage the authors to discuss this phenomenon in the paper.

Minor edits:

Line 23 – “suitable to a limited extend”  should be “limited extent”

Line 47 – “the system thrives”  should be “the system strives”

Line 161 – “the only exemptions”  should be “the only exceptions”

Round 2

Reviewer 2 Report

I very much appreciate the authors’ efforts to address my original concerns. Although it is disappointing that the authors aren’t able to include data with larger pore size membranes or in phosphate buffer, the revised version of the paper does provide a useful contribution to the field.  I am pleased to recommend publication in its current form.